# Implementing Defects for Ratiometric Luminescence Thermometry

**DOI:** 10.3390/nano10071333

**Published:** 2020-07-08

**Authors:** Joanna Drabik, Karolina Ledwa, Łukasz Marciniak

**Affiliations:** Institute of Low Temperature and Structure Research, Polish Academy of Sciences, Okólna 2, 50-422 Wroclaw, Poland; k.ledwa@intibs.pl

**Keywords:** defects, lanthanide oxide, terbium, nanothermometer, luminescent thermometry, phosphor

## Abstract

In luminescence thermometry enabling temperature reading at a distance, an important challenge is to propose new solutions that open measuring and material possibilities. Responding to these needs, in the nanocrystalline phosphors of yttrium oxide Y_2_O_3_ and lutetium oxide Lu_2_O_3_, temperature-dependent emission of trivalent terbium Tb^3+^ dopant ions was recorded at the excitation wavelength 266 nm. The signal of intensity decreasing with temperature was monitored in the range corresponding to the ^5^D_4_ → ^7^F_6_ emission band. On the other hand, defect emission intensity obtained upon 543 nm excitation increases significantly at elevated temperatures. The opposite thermal monotonicity of these two signals in the same spectral range enabled development of the single band ratiometric luminescent thermometer of as high a relative sensitivity as 4.92%/°C and 2%/°C for Y_2_O_3_:Tb^3+^ and Lu_2_O_3_:Tb^3+^ nanocrystals, respectively. This study presents the first report on luminescent thermometry using defect emission in inorganic phosphors.

## 1. Introduction

A whole range of optically active emitters can be successfully used in luminescence thermometry due to the susceptibility of their optical properties to the temperature changes, like: organic dyes, quantum dots (QDs), metal–organic frameworks (MOFs), polymers, nanodiamonds (ND), inorganic nanoparticles doped with lanthanides (Ln^3+^) and transition metal ions (TM) etc. Organic dyes [1,2,3,4] usually emit in the visible range upon excitation with ultraviolet light. Their emission is dependent on temperature, because it is affected by the electronic transitions between vibrational states responsible for luminescence. The luminescent properties of QDs, which result from the recombination of the electron–hole pair are strongly dependent both on the host material composition as well as their size [5,6,7]. In this case, the phonon-related depopulation processes and the energy gap are strongly temperature dependent. Therefore, QDs are often implemented to spectral position- and intensity-based thermometric approaches. However, their use as the source of one of the luminescent signals in more complex ratiometric nanothermometric systems can also be found [8,9]. Luminescent thermometers (LTs) based on MOFs take advantage from the fact that probability of host-to-metal and metal-to-metal energy transfer probabilities are thermally induced, to enable noncontact temperature readout [10,11,12,13]. The unlimited number of combinations of various ions and links, enables designing structures with intentional properties. Polymers [14,15,16,17,18], which most often reveal visible luminescence upon UV excitation, are another group of materials useful in LTs. In spite of their relatively low quantum efficiency, they constitute a significant group of nanothermometers due to their good solubility in water and the strong dependence of their luminescence on local environment changes. By combining polymers, for example with organic dyes or QDs, it is possible to obtain very sensitive sensors that use polymer conformational changes. In luminescence thermometry, examples of the use of color centers found in diamonds, such as luminescent tin-vacancy [19], have also been noted. Thermally induced shifting of the zero-phonon line as well as a change in the intensity of the emission of these intentionally introduced optically active centers was observed.

Interesting alternatives to already described LTs pose inorganic nanocrystals doped with optically active ions, most often lanthanides [20] or transition metals [21,22,23,24]. A number of their advantages, over other groups of material used for LTs, like high chemical and thermal stability and the lack of photobleaching and photoblinking, lead to the great research interest that they have attracted. In this case, the difference in the thermal dependence of particular emission bands enables the implementation of the ratiometric approach. The optimization of the host material composition and the optically active ions enables the alteration of the thermometric properties of this type of luminescent thermometers according to the requirements. Due to the fact that each of those already reported types of LTs possess some drawbacks and there is no universal luminescent temperature sensor, new approaches and new materials that may be implemented to noncontact temperature sensors are still searched for.

In the nanocrystalline lanthanide oxides, broad bands from surface oxygen defects have been noted. They were investigated in e.g., Y_2_O_3_ doped with Eu^3+^ [25] or Tb^3+^ [26], in which the defect emission intensity was comparable to that of dopant ions in a certain temperature range. The influence of temperature on the emission of defects has been demonstrated, which led us to the conclusion that such structures could be intentionally used in optical thermometry. Usually, the emission of such structures is quenched with temperature, however, for energy mismatches of excitation radiation, it is possible to note an increase in the intensity of defect emission, which is associated with satellite phonons [25]. In this work, we show for the first time the exploitation of emission of oxygen defects, occurring in rare earth oxides, in excitation-ratiometric approach to luminescence thermometry using Tb^3+^ ions emission as the luminescent reference. It has been shown that the Tb^3+^ emission band obtained upon UV excitation decreases at elevated temperatures. On the other hand, the defect emission upon green excitation reveals thermal enhancement. The opposite monotonicity of these two signals integrated in the same spectral range enables the development of single band ratiometric (SBR) luminescent thermometers based on intrinsic defect emission (Figure 1).

## 2. Materials and Methods

The nanocrystalline rare earth (RE) oxides phosphors (Y_2_O_3_ and Lu_2_O_3_) doped with the trivalent terbium Tb^3+^ ions (0.05%, 0.1%, 0.2% and 0.5% mole in respect to Y^3+^ and Lu^3+^, respectively) were synthesized using the citric acid-assisted method. The following compounds were used as starting materials: yttrium oxide (Y_2_O_3_ of 99.999% purity from Stanford Materials Corporation (SMC, Lake Forest, CA, USA.), lutetium oxide (Lu_2_O_3_ of 99.995% purity from SMC), terbium oxide (Tb_4_O_7_ of 99.99% purity from SMC), nitric acid (HNO_3_ of 65% purity from Avantor, Gliwice, Slaskie, Poland), and citric acid (C_6_H_8_O_7_ of 99% purity from Sigma-Aldrich, Poznan, Wielkopolskie, Poland). The amounts of all the precursors used in the synthesis were stoichiometrically calculated per 0.5 g of product. The first step was the production of the rare earth nitrates by addition of a significant excess of HNO_3_ to the water solutions of the oxides. Then, the residual HNO_3_ was eliminated by a recrystallization process. The citric acid was then added, and the mixture was kept at the heating plate at around 100 °C in ambient atmosphere for about a day to obtain a resin. The ratio of citric acid moles to metal moles was 1:2. The resultant substance was annealed at 900 °C for 8 h in air.

Powder diffraction studies were carried out on a PANalytical X’Pert Pro diffractometer equipped with an Anton Paar TCU 1000 N temperature control unit using Ni-filtered Cu Kα radiation (V = 40 kV, I = 30 mA). Rietveld’s analysis was performed using the X’Pert HighScore Plus software (version 2.2d (2.2.4), Malvern Panalytical, Malvern, Worecestershire, UK).

Raman spectra were measured on inVia confocal microscope from Renishaw supplied with the Si CCD camera for detection and an 830 nm excitation line. The spectra were taken at room temperature under 20× objective. The spatial resolution was lower than 1 µm.

Transmission electron microscope images were taken using the FEI Tecnai G2 20 X-TWIN microscope supplied with the CCD FEI Eagle 2K camera with a HAADF detector and electron gun with a LaB6 cathode. Moreover, the microscope was supplied with the X-ray microanalyzer EDAX.

The excitation spectra, the photoluminescence decay curves and the quantum yield measurements were measured at room temperature using the FLS980 fluorescence spectrometer from Edinburgh Instruments (version 980, Edinburgh Instruments, Livingston, UK). The measurements were carried out with R928P side window photomultiplier tube from Hamamatsu detector and an integrating sphere for quantum yield measurements. The excitation line was obtained using a 450 W halogen lamp and the micro-flash lamp.

The temperature-dependent emission spectra were measured using the 266 nm excitation line from a laser diode or the 543 nm line from OPOLLETE 355 LD OPO and measured using a Silver-Nova Super Range TEC Spectrometer form Stellarnet (1 nm spectral resolution, Tampa, FL, USA.). The temperature of the sample was controlled using a THMS 600 heating stage from Linkam (0.1 °C temperature stability and 0.1 °C set point resolution). Measurements of emission over a wide temperature range with both of the excitation lines were carried out using band pass filters 400–500 nm from Thorlabs. The temperature was reduced from 200 °C to 20 °C with 20 °C steps, whereas the time needed to stabilize the temperature controller between the double emission measurements was about 2 min.

## 3. Results

The obtained RE oxide nanocrystals underwent structural and morphological characterization. As for the crystal structure, all the studied materials crystallize in the space group I a −3 (206) with a cubic unit cell. Rare earth atoms located inside the ordered volume of nanocrystals are placed in the octahedral sites, having six atoms of oxygen as their nearest neighbors. The X-ray powder diffraction results are shown in Figure 2a. An adequate phase purity was confirmed by the consistency of the received diffractograms with the ICSD pattern data (#193042, #428543 for Y_2_O_3_ and Lu_2_O_3_, respectively). No shifts or changes in the relative peak intensities were observed for the increase in the Tb^3+^ dopant concentration. This is because the Tb^3+^ ion easily and without a noticeable impact substitutes for Y^3+^ and Lu^3+^ ions, due to the fact that those ions have similar ionic radii: 106.3 pm, 104.0 pm and 100.1 pm, respectively [27]. Based on the Rietveld refinement with respect to the reference data, the average sizes of the obtained nanocrystals were estimated (Appendix A). Despite the uncertainty of such analysis associated with the probably significant size distribution, some trend is visible: Y_2_O_3_ nanocrystals are larger than Lu_2_O_3_, which agrees with the values of RE ionic radii. For larger ions, a larger unit cell is obtained (see Appendix A), which can result in such a dependence of crystalline sizes. In fact, a smaller unit cell parameter was determined for Lu_2_O_3_ than for Y_2_O_3_, which were about 1.039 nm and 1.061 nm, respectively. Only a slight increase in this parameter was recorded in both nanocrystalline matrices when the Tb^3+^ content was increased. However, due to the chosen synthesis method, the surface of nanocrystals is highly defective. The value of the shortest metal–oxygen distances (M–O) is also larger in the case of Y_2_O_3_ (0.2206 nm) with respect to the Lu_2_O_3_ (0.2189 nm), which indicates that stronger crystal field strength affects the dopant ion in Lu_2_O_3_ (Figure 2b) [28]. Because of slight changes in the electron–phonon coupling, different phonon energies were observed in Raman spectra (Figure 2c). In both RE oxides, the arrangement of the peaks is similar and the appearing modes are marked on the graph (A-singly degenerated, E-doubly degenerated, T-triply degenerated, g-even) [29]. In the case of Y_2_O_3_, the corresponding peaks are shifted towards the lower energies with respect to Lu_2_O_3_, which results from the smaller unit cell of the Lu_2_O_3_ host. The energy of the dominant Raman peak is observed around 378 for Y_2_O_3_ while around 392 cm^−1^ for Lu_2_O_3_. Moreover, its width increases slightly for Lu_2_O_3_ relative to Y_2_O_3_, which is consistent with the tendency shown by the average size of crystallites (Appendix A), because the smaller the grains, the larger the peak width. The analysis of the TEM images indicates that Y_2_O_3_:Tb^3+^ and Lu_2_O_3_:Tb^3+^ nanocrystalline powders consist of well crystalized and highly agglomerated particles of the average size of around 56 and 52 nm, respectively (Figure 2d, see also Appendix A). Obtained results are also consistent with average crystallite sizes calculated from XRD data.

The optical properties of the studied nanocrystallites were also investigated. The excitation spectra measured at λ_em_ = 540 nm matching ^5^D_4_ → ^7^F_5_ transition is presented in Figure 3b. In the case of both RE oxides, two broad bands in the ultraviolet range are visible. The first one around 260 nm corresponds to the spin-allowed 4*f*^8^
→ 4*f*^7^5*d*^1^ transition of the Tb^3+^ ion, while the band around 300 nm corresponds to the spin-forbidden 4*f* → 5*d* transition [30]. Because of the large width of the observed bands, and the fact that the measurements were carried out at room temperature, it is not possible to accurately determine whether the absorption resulting from the presence of defects is visible in the excitation spectrum. For both RE oxides, an increase in the relative intensity of the excitation bands of Tb^3+^ ions was observed for higher concentrations of Tb^3+^ ions in the crystalline matrix. This is due to the fact that a larger amount of Tb^3+^ ions results in the increase of the absorption cross-section, which in turn causes stronger emission. Due to the relatively large energy separation between ^5^D_4_ and next lower laying energy state (^7^F_0_) and the distinctive energy level scheme of Tb^3+^ ions, the ^5^D_4_ emitting state is barely affected by either multiphonon depopulation or cross relaxation processes. Therefore, an increase of the dopant concentration results in the enhancement of the emission intensity. The representative emission spectra measured at λ_exc_ = 266 nm for RE oxides with 0.5% Tb^3+^ shown in Figure 3c consist of a series of bands around 490, 544, 584, 623, 650, 666 and 685 nm that correspond to the transitions from the ^5^D_4_ level to the ^7^F_J_ sublevels (J = 6, 5, 4, 3, 2, 1 and 0, respectively). Although the emission spectra of Tb^3+^ doped RE oxides are very similar qualitatively, there is a large quantitative difference in the absolute brightness of emissions. Upon 266 nm resonant excitation, stronger emission intensity is observed for Y_2_O_3_:Tb^3+^, which is confirmed by quantum yield measurements presented in Appendix A. Moreover, with the increase of Tb^3+^ ion concentration, the quantum yield increases for each RE oxide. This effect is related to the fact that, in the case of nanocrystals, Ln^3+^ ions prefer to occupy defective surface positions, which are strongly quenched by the surface-related nonradiative processes [31]. The increase of Tb^3+^ concentration leads to the higher probability of less surface-affected sites’ occupation in the core part of the nanocrystals. Additionally, as was already mentioned, the concentration quenching is not very probable in the case of the ^5^D_4_ state. Synergy of these two effects causes the enhancement of emission intensity and luminescence quantum yield with enlargement of the Tb^3+^ amount. It is therefore evidenced that the surface is strongly defected in the case of the investigated nanocrystals. The obtained luminescence decay curves are presented in Figure 3d for the representative 0.5% of Tb^3+^ concentration. The lifetime elongation of the ^5^D_4_ energy state of Tb^3+^ ions in the Lu_2_O_3_, with respect to Y_2_O_3_, is associated with the higher spin-orbital coupling in the case of Lu_2_O_3_ nanocrystals. For the lowest Tb^3+^ concentration under investigation, the lifetimes of 0.90 ms and 0.93 ms were obtained, for Y_2_O_3_ and Lu_2_O_3_, respectively. For a concentration of 0.5% Tb^3+^, for which the luminescence decay curves are presented in Figure 3d, the obtained lifetime values were 0.86 for Y_2_O_3_ and 0.89 for Lu_2_O_3_. Decay times also show a dependence on Tb^3+^ concentration. The lifetimes determined by single-exponent fit are shown in Appendix A. It was observed that for the tenfold increase in concentration of Tb^3+^, shortening of average lifetimes reached only 4–7% of the initial value obtained for the lowest dopant concentration. For Y_2_O_3_, the change from 0.90 ms to 0.86 ms was noted for the increase in Tb^3+^ content in the matrix from 0.05% to 0.5%, whereas for Lu_2_O_3_ a change from 0.93 ms to 0.89 ms was observed. The shortening of lifetimes as the Tb^3+^ concentration increases results from the fact that the average distance between ions decreases, which in turn opens the paths of energy migration among excited states of Tb^3+^ ions to the surface quenchers. However, in this case, the shortening of the lifetimes for higher Tb^3+^ concentrations have no direct effect on the intensity of emissions, because the competitive process associated with the increase in the number of emitters located in the nanocrystal volume is dominant. That is why in the excitation spectrum the intensity of the emission band at 540 nm increases for higher Tb^3+^ concentrations (Figure 3b).

The luminescent properties of the obtained RE oxides nanocrystals doped with Tb^3+^ ions were investigated in a wide temperature range from 20 to 200 °C upon λ_exc_ = 266 nm and λ_exc_ = 543 nm and an appropriate band pass 400–500 nm filter. It was estimated from the excitation spectra that the 266 nm excitation line is suitable for the direct resonant excitation of Tb^3+^ ions. This ultraviolet line causes the 4*f* → 5*d* transitions of Tb^3+^ ions, and then leads to the population of ^5^D_3_ and ^5^D_4_ levels from which the emission occurs. However, the emission from the ^5^D_3_ level is negligible, which may result from the probable {^5^D_4_, ^7^F_6_}↔{^5^D_3_, ^7^F_0_} cross-relaxation processes. The representative thermal evolution of the Y_2_O_3_:0.5%Tb^3+^ emission spectra in the monitored spectral range is shown in Figure 4a. Only one emission band associated with the ^5^D_4_ → ^7^F_6_ transition is observed in this spectral range, the intensity of which decreases with increasing temperature. This effect may be associated with either multiphonon depopulation of the ^5^D_4_ state, of which probability is dependent on temperature, or with a decrease in the absorption cross-section as the temperature increases resulting from the increase in electron–phonon coupling [32]. However, due to the large energy separation between ^5^D_4_ and ^7^F_6_ states and the required number of phonons to bridge this gap, the thermally-induced enhancement of multiphonon nonradiative processes is not sufficiently efficient to explain the observed thermal dependence. Therefore, as it was proved in the course of our previous studies [32], the temperature-dependent change of the absorption cross section is the dominant process responsible for the lowering of the ^5^D_4_ → ^7^F_J_ emission of Tb^3+^ ions. On the other hand, when an excitation wavelength was used that was not in resonance with the absorption from the ground state of Tb^3+^, a peculiar broadband emission was found (Figure 4b). The intensity of this emission band cut-off by the used shortpass optical filter increases at elevated temperatures. To verify the source of this broadband emission, a similar measurement was performed on the Y_2_O_3_ microcrystalline powder. The performed studies indicate that the similar shape of emission bands was noticed in this case, however of significantly lower intensity (Appendix A). This excluded the contribution of Tb^3+^ ions in the generation of broadband emission. As reported previously, RE oxides exhibit intrinsic luminescent properties associated with optically active defects [25,26]. Huang et al. [33] reported that, in the case of nanocrystalline Y_2_O_3_, the peak observed in the X-ray photoelectron spectroscopy (XPS) corresponding to binding energy of O1s was significantly broader with respect to those observed for their microcrystalline counterpart. This clearly indicated the presence of different chemical states of O1s ions in the Y_2_O_3_ nanocrystals related to the defect states. Broad defect emission was previously reported for other oxides i.e., ZnO, for which a heterogeneously broadened band ranging from 400 to 600 nm was reported by Karthikeyan et al. [34]. Foch et al. reported that the position of the defects band maximum in Si+ implemented SiO_2_ can be altered by the preparation condition. The enhancement of the defect emission intensity in the Y_2_O_3_ and Lu_2_O_3_ nanocrystals with respect to the microcrystalline counterpart may suggest that mainly surface defects are involved in this process. Interestingly the emission intensity of defect states increases with temperature. This is probably associated with the fact that excitation wavelength reaches the sideband of the defects absorption band and, at an elevated temperature, the broadening of the absorption band leads to the more efficient pumping of defect emission. Confirmation of this hypothesis is the fact that in the case of the Y_2_O_3_ microcrystals, for which the presence of much narrower absorption band is expected, defect emission can be observed only above 140 °C. The abovementioned emission characteristics for each of the used laser lines were tested for both RE oxide matrices and for different concentrations of Tb^3+^. The Tb^3+^ emission intensity obtained upon λ_exc_ = 266 nm was integrated in the range corresponding to the ^5^D_4_ → ^7^F_6_ band and its temperature dependence was depicted in Figure 4c. There is a gradual decrease in the intensity of emission when the temperature increases from 20 to 200 °C: for Y_2_O_3_, a decrease to about 60% of the initial value was recorded, while for Lu_2_O_3_, this decrease is greater, to about 40–50%, which is related to the crystal field strength. On the other hand, upon λ_exc_ = 543 nm, significantly, over 5-fold enhancement of defect emission intensity at elevated temperature was observed (Figure 4d), while for the Y_2_O_3_:0.05%Tb^3+^ nanocrystals even 10-fold enhancement of emission intensity was found. The opposite thermal monotonicity of two signals: (1) luminescence of Tb^3+^ ions and (2) defect emission, which occur in the same spectral range upon different λ_exc_, enables development of the SBR luminescent thermometers based on Y_2_O_3_:Tb^3+^ and Lu_2_O_3_:Tb^3+^ nanocrystals. In order to verify the performance of nanocrystals under investigation to noncontact temperature sensing, the ratio of integral luminescence intensities (LIR) upon λ_exc_ = 266 nm and λ_exc_ = 543 nm calculated in the same spectral range was determined:(1)$$LIR=\frac{I\left[Tb^{3+}\right]}{I\left[defects\right]}=\frac{\int_{400nm}^{500nm}I{\left(T\right)}_{\lambda exc=266nm}d\lambda }{\int_{400nm}^{500nm}I{\left(T\right)}_{\lambda exc=543nm}d\lambda }$$

As shown in Figure 4e, a 12-fold enhancement of the LIR for Y_2_O_3_:Tb^3+^ nanocrystals can be observed in the 40–200 °C temperature range. In the case of the Lu_2_O_3_:Tb^3+^ nanocrystals observed, LIR’s thermal enhancement was only slightly lower due to the fact that although the thermally induced defect emission intensity was significantly lower with respect to the Y_2_O_3_:Tb^3+^ counterpart, the Tb^3+^ emission intensity was quenched faster in this case. The quantitative analysis of the thermally induced LIR’s changes is provided by the calculation of the relative sensitivity (S_r_) of SBR LT to temperature changes defined as follows:(2)$$S_{r}=\frac{1}{LIR}\frac{\Delta LIR}{\Delta T}\cdot 100\%$$
where Δ*LIR* represents the change of *LIR* corresponding to Δ*T* change of temperature. The maps presenting the thermal dependence of *S_r_* for different dopant concentration shown in Figure 4g indicate that in the case of Y_2_O_3_:Tb^3+^ nanocrystals, there is no correlation between dopant concentration and the S_r_. The highest *S_r_* values were found in the 60–100 °C temperature range, for which with the maximum *S_r_* = 4.92%/°C at 40 °C for Y_2_O_3_:0.05%Tb^3+^. On the other hand, in the case of Lu_2_O_3_:Tb^3+^ nanocrystals, the increase of dopant concentration leads to the gradual enhancement of the *S_r_* value. This effect may result from the difference in the ionic radii of Lu^3+^ and Tb^3+^. The increase of the dopant ion concentration of larger ionic radii with respect to the substituted one results in the generation of the larger number of defect states. The analogous difference in the ionic radii in the case of Y_2_O_3_:Tb^3+^ nanocrystals is only slight.

## 4. Conclusions

In this work, to the best of our knowledge, for the first-time defect emission of oxide nanoparticles was implemented to noncontact temperature sensing. In order to determine the usefulness for luminescence thermometry, the spectroscopic properties of Y_2_O_3_:Tb^3+^ and Lu_2_O_3_:Tb^3+^ nanoparticles were examined in a wide temperature range. In was found that the emission intensity of Tb^3+^ bands associated with the ^5^D_4_ → ^7^F_J_ electronic transition obtained upon λ_exc_ = 266 nm decreases with temperature. This thermal quenching process was explained in terms of lowered absorption cross section at the excitation wavelength at elevated temperatures. On the other hand, emission intensity of the defect states increases significantly by one order of magnitude in the temperature range under investigation. The comparative studies with the micro sized counterparts reveal the dominant contribution of the surface states in the generation of the emission. The opposite thermal dependence of these two independently excited luminescent signals enabled the development of a single band ratiometric luminescent thermometer. The high thermometric performance of the presented approach was confirmed by the high S_r_ values as 4.92%/°C for Y_2_O_3_:0.05%Tb^3+^ and 2%/°C for Lu_2_O_3_:0.5%Tb^3+^ nanocrystals. Additionally, it was found that the increase of the Tb^3+^ concentration in Lu_2_O_3_ nanocrystals enhances the S_r_ values that were discussed in terms of higher probability of the defect creation in this host material associated with the difference in the ionic radii between Lu^3+^ and Tb^3+^ ions. The use of this new type of optically active center opens up a number of new possibilities and the high relative sensitivities achieved indicate that this achievement should not be overlooked and ought to be further developed.

## Figures and Tables

**Figure 1 nanomaterials-10-01333-f001:**
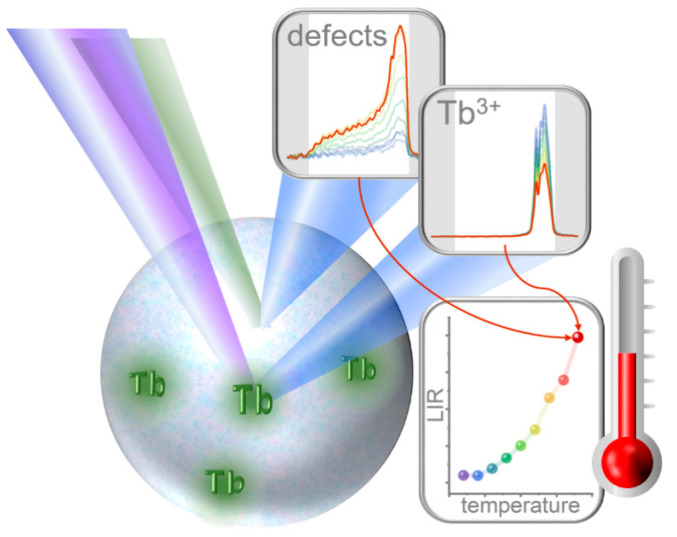
Visualization of the applied measurement scheme: two wavelengths of 266 nm and 543 nm exciting Tb^3+^ ions and luminescent defects, respectively, enabling the local temperature readout on the basis of the luminescence intensity ratio (LIR).

**Figure 2 nanomaterials-10-01333-f002:**
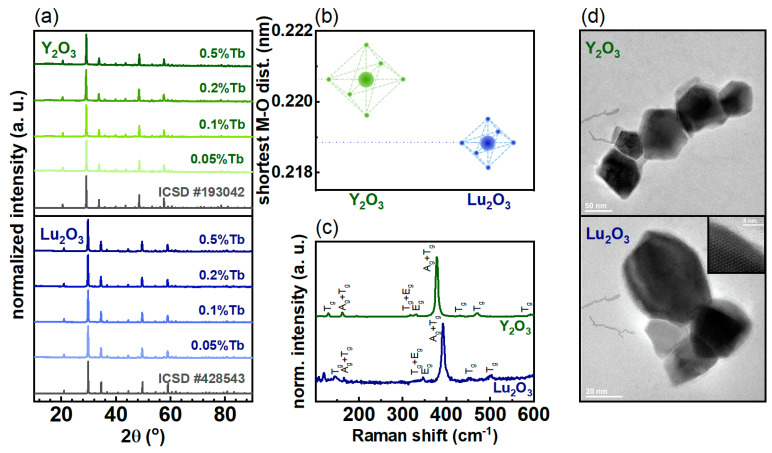
(**a**) X-ray diffraction patterns of the nanocrystalline rare earth (RE) oxides doped with different concentrations of the Tb^3+^ ions; (**b**) Shortest metal–oxygen distances in the given crystals; (**c**) Raman shift; (**d**) Transmission Electron Microscopy images of the nanocrystalline RE oxides doped with 0.5% Tb^3+^ ions (the appropriate scale bar is presented in each image).

**Figure 3 nanomaterials-10-01333-f003:**
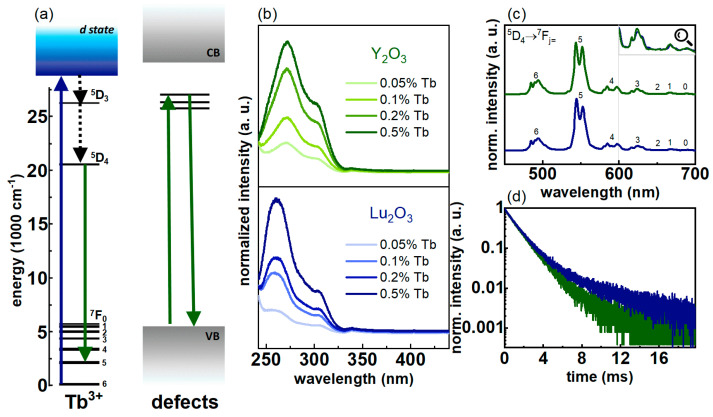
(**a**) Energy schemes of the luminescent centres present in the investigated RE oxides doped with Tb^3+^; (**b**) Excitation spectra (λ_em_ = 540 nm) of the RE oxides doped with different concentrations of the Tb^3+^ ions (normalized to the last obtained value); (**c**) Emission spectra (λ_exc_ = 266 nm) measured at room temperature for the RE oxides doped with 0.5% Tb^3+^; (**d**) Luminescence decay curves (λ_exc_ = 375 nm, λ_em_ = 540 nm).

**Figure 4 nanomaterials-10-01333-f004:**
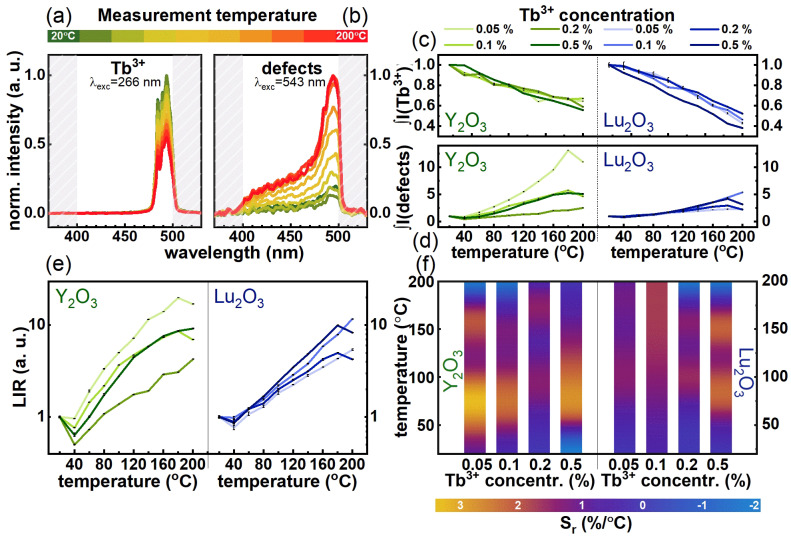
(**a**). Representative thermal evolution of the emission spectra for Y2O3:0.5% Tb3+ under 266 nm and (**b**) under 543 nm excitation; (**c**) Thermal evolution of the integrated RE oxides doped with different Tb3+ ions emission intensity under 266 nm and (**d**) 543 nm excitation; (**e**) Luminescent intensity ratio; (**f**) Relative sensitivity.

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
