# Peer review of "Implementing Defects for Ratiometric Luminescence Thermometry"

_nanomaterials, 2020, doi:10.3390/nano10071333_

Round 1

Reviewer 1 Report

Peer-review of Manuscript nanomaterials-851864

"Implementing defects for ratiometric luminescence thermometry" by Joanna Drabik and Lukasz Marciniak

This interesting paper about new solutions for luminescence thermometry based on nanocrystalline phosphors
of yttrium oxide and lutetium should be published after the following minor revisions.

line 44, ...conformational changes there is a note with number 8, it is not clear what it is referring to
lines 82-85 for materials purity please insert a refeence to data-sheets
lines 102-103 "Linkam (0.1 °C temperature stability and 0.1 °C set point resolution)." these values are from the data-sheet? does it mean that the stability is at level of instrument sensitivity?
line 105 "every 20 °C" could it be changed with "with 20 °C steps"
line 217 refs. 25,26 are called in a different style
line 223 Foch et al. reference missing
line 234 5D4 7F6 symbol missing
line 249 and 260 in respect -> with respect to

I suggest to split figures 3 and 4 panels in more figures to help the reader
Figure 3b missing, may be that b is c, c is d and d is e?

About supplementary pictures, In my opinioon the ones which are called in the text should be added in the paper or you should add a reference containing them

Author Response

Authors reply:

Thank you for the positive opinion and for the constructive comments to which we refer below.

  1. line 44, ...conformational changes there is a note with number 8, it is not clear what it is referring to

Authors reply:

Thank you for bringing this to our attention. Number 8 is a misplaced reference. It has been removed in the revised version of the manuscript.

  1. lines 82-85 for materials purity please insert a refeence to data-sheets

Authors reply:

Thank you for this remark. Unfortunately, manufacturers do not offer references to quote. For the Reviewer's insight we attach the safety data sheets. Unfortunately, no safety data sheet was found for Tb4O7, and for HNO3 the data sheet is in Polish:

  • Y2O3 (no. OX39-5N): https://www.samaterials.com/content/96-msds-of-yttrium-oxide
  • Lu2O3 (no. OX71-4N5): https://www.samaterials.com/pdf/Lutetium-Oxide-(Lu2O3)-sds.pdf
  • Tb4O7 (no. OX65-4N): no data sheet
  • HNO3: http://www.poch.com.pl/wysw/utworz_pdf.php?nr_karty=586 (in Polish)
  • C6H8O7: https://pubchem.ncbi.nlm.nih.gov/substance/329774820
  1. lines 102-103 "Linkam (0.1 °C temperature stability and 0.1 °C set point resolution)." these values are from the data-sheet? does it mean that the stability is at level of instrument sensitivity?

Authors reply:

Thank you for this question. These parameters are from the apparatus description. According to the data both stability and resolution parameters are equal.

  1. line 105 "every 20 °C" could it be changed with "with 20 °C steps"

Authors reply:

Thank you for this suggestion. It has been included in the revised version of the manuscript.

  1. line 217 refs. 25,26 are called in a different style

Authors reply:

Thank you for bringing this to our attention. This error is corrected in the revised version of the manuscript.

  1. line 223 Foch et al. reference missing

Authors reply:

Thank you for bringing our attention to this omission. The missing reference has been added to the revised version of the manuscript.

  1. line 234 5D4 7F6 symbol missing

Authors reply:

Thank you for this remark. The lacking arrow was included in the revised version of the manuscript.

  1. line 249 and 260 in respect -> with respect to

Authors reply:

We are grateful for this remark. The wrong preposition has been replaced with a correct one.

  1. I suggest to split figures 3 and 4 panels in more figures to help the reader

Authors reply:

Thank you for this suggestion. However, the choice of such panels in the Figures was dictated by the desire to collect data on one issue in one place so that the Reader does not get lost. Figure 3 contains information about spectroscopic properties, while Figure 4 shows all thermometry data. Separating them, in our opinion, will not be beneficial to the perception of this work.

  1. Figure 3b missing, may be that b is c, c is d and d is e?

Authors reply:

Thank you for noticing this oversight. Indeed, mark (b) has been omitted, and subsequent letters have been used incorrectly. The order suggested by the reviewer is correct. The change has been included in the revised manuscript.

  1. About supplementary pictures, In my opinioon the ones which are called in the text should be added in the paper or you should add a reference containing them

Authors reply:

Thank you for this proposal. In our opinion, the Figures contained in Supporting Information are not significant enough to be included in the manuscript. However, for an inquisitive Reader they can be helpful, which is why they are included in the Supporting Information. These are our own results, so adding a reference is not possible.

Reviewer 2 Report

This is a nice work by Marciniak et al, with relevant scientific findings to the lanthanide thermometry community. The work is carried out well and the manuscript is prepared in a concise and clear way. I recommend this manuscript for publication in its current form. 

Author Response

Authors reply:

We express gratitude for the positive opinion of the Reviewer.

Reviewer 3 Report

Review of the article “Implementing defects for ratiometric luminescence thermometry” by J. Drabik and L. Marciniak, Nanomaterials, Manuscript ID nanomaterials-851864.

In this article, nanophosphors based on Y2O3 and Lu2O3 doped with Tb3+ ions are prepared and photophysically characterized. The authors propose such nanosystems as single band ratiometric luminescent thermometers, which present a different luminescence response in the same spectral region (400-500 nm) depending on the used excitation wavelength. The increase of temperature induces a decrease in the luminescence intensity of the 5D4 7F6 transition upon laser excitation at 266 nm, and an increase of the luminescence intensity of the band ascribable to oxygen defects in the crystal structure of the nanophosphors upon laser excitation at 543 nm. A relative sensitivity of 4.92 %/°C for Y2O3:Tb3+ and 2 %/°C for Lu2O3:Tb3+ is registered.

Major comments:

  1. The article is well organized and clear. Nevertheless, the authors did not insert important synthetical and morphological data.
  2. The synthetic procedure (lines 79-88) of the nanophosphors studied by the authors is not exhaustive. The authors should provide details about the amounts of used reagents and solvents, reaction temperatures and durations, use of inert atmosphere, reagents used in purification steps et cetera.
  3. The morphological characterization of the prepared nanomaterial is incomplete. TEM images of the nanomaterials have to be added to demonstrate the presence of nanometric primary particles, and the relevant particle size distribution analysis has to be performed and discussed.
  4. The method used to register luminescence quantum yields is not described in section Materials and Methods. The authors should add the description of the procedure used for the assessment of luminescence quantum yields.
  5. Lines 172-173: please insert the discussion about the lifetimes calculated for the 0.5% Tb3+-doped sample, to be consistent with the previous text and with Figure 3(d).
  6. The letter assignations in Figure 3 are not correct: (c) is (b), (d) is (c), (e) is (d).
  7. Line 198: the correct cross relaxation is {5D4, 7F6} ↔ {5D3, 7F0}.

Minor comments:

  1. Line 86: I would change the sentence to “the residual HNO3 was eliminated…”
  2. Line 117, after “respectively”: please add a reference.
  3. Line 145, after “4f→5d transition”: please add a reference.
  4. Line 153: change “barley” to “barely”.
  5. Line 171: change to “with respect to”.
  6. Line 204, after “phonon coupling”: please add a reference.
  7. Line 207, after “previous studies”: please add a reference.
  8. Line 234: missing arrow in 5D47F6
  9. Line 254: change to “Figure 4(g)”.
  10. Line 270: verb is missing after “in order to”.
  11. Figure S4: the legend is missing.

Author Response

Authors reply:

We are thankful for all the comments sent by the Reviewer. They constitute a great contribution to our work.

Major comments:

  1. The article is well organized and clear. Nevertheless, the authors did not insert important synthetical and morphological data.

Authors reply:

Thank you for this remark. The synthesis description has been extended. It contains more important details. In addition, TEM images have been added in the revised manuscript.

  1. The synthetic procedure (lines 79-88) of the nanophosphors studied by the authors is not exhaustive. The authors should provide details about the amounts of used reagents and solvents, reaction temperatures and durations, use of inert atmosphere, reagents used in purification steps et cetera.

Authors reply:

Thank you for listing these important parameters. They were all added to the synthesis description, as recommended by the Reviewer.

  1. The morphological characterization of the prepared nanomaterial is incomplete. TEM images of the nanomaterials have to be added to demonstrate the presence of nanometric primary particles, and the relevant particle size distribution analysis has to be performed and discussed.

Authors reply:

Thank you for this attention. As recommended by the reviewer, TEM images have been added to the revised version of the manuscript (Figure 2d) and to the revised Supporting Information (Figure S3). The appropriate passage has been added to the text of the revised manuscript.

  1. The method used to register luminescence quantum yields is not described in section Materials and Methods. The authors should add the description of the procedure used for the assessment of luminescence quantum yields.

Authors reply:

Thank you for bringing this to our attention. This is a major oversight on our part, thus the corrected version of the manuscript includes a description of the procedure for measuring quantum efficiency in the Methods section.

  1. Lines 172-173: please insert the discussion about the lifetimes calculated for the 0.5% Tb3+-doped sample, to be consistent with the previous text and with Figure 3(d).

Authors reply:

Thank you for this attention. Discussion of the lifetimes obtained for the concentration of 0.5% Tb3+ was added to the revised version of the manuscript.

  1. The letter assignations in Figure 3 are not correct: (c) is (b), (d) is (c), (e) is (d).

Authors reply:

Thank you for noticing this oversight. Indeed, mark (b) has been omitted, and subsequent letters have been used incorrectly. The order suggested by the reviewer is correct. The change has been included in the revised manuscript.

  1. Line 198: the correct cross relaxation is {5D4, 7F6} {5D3, 7F0}.

Authors reply:

Thank you for this remark. The incorrect cross-relaxation notation have been corrected in the revised manuscript.

Minor comments:

  1. Line 86: I would change the sentence to “the residual HNO3 was eliminated…”

Authors reply:

Thank you for this remark. The change suggested by the reviewer was introduced to the text of the revised manuscript.

  1. Line 117, after “respectively”: please add a reference.

Authors reply:

We are grateful for this remark. The appropriate reference has been added in the mentioned sentence.

  1. Line 145, after “4f5d transition: please add a reference.

Authors reply:

Thank you for this remark. The appropriate reference has been added in the mentioned sentence.

  1. Line 153: change “barley” to “barely”.

Authors reply:

Thank you for pointing out this error. It has been corrected in the revised version of manuscript.

  1. Line 171: change to “with respect to”.

Authors reply:

We are grateful for this remark. The wrong preposition has been replaced with a correct one.

  1. Line 204, after “phonon coupling”: please add a reference.

Authors reply:

We are thankful for this remark. The appropriate reference has been added in the mentioned sentence.

  1. Line 207, after “previous studies”: please add a reference.

Authors reply:

Thank you for this remark. The appropriate reference has been added in the mentioned sentence.

  1. Line 234: missing arrow in 5D47F6

Authors reply:

Thank you for this remark. The lacking arrow was included in the revised version of the manuscript.

  1. Line 254: change to “Figure 4(g)”.

Authors reply:

Thank you for bringing this to our attention. The wrong panel marking has been corrected.

  1. Line 270: verb is missing after “in order to”.

Authors reply:

Thank you for this remark. The missing part of the sentence has been completed in the revised manuscript.

  1. Figure S4: the legend is missing.

Authors reply:

Thank you for bringing this to our attention. A missing legend has been added.

Round 2

Reviewer 3 Report

Second review of the article “Implementing defects for ratiometric luminescence thermometry” by J. Drabik and L. Marciniak, Nanomaterials, Manuscript ID nanomaterials-851864.

The authors replied satisfactorily to all of the questions and comments.